# Fine-Scale Space-Time Cluster Detection of COVID-19 in Mainland China Using Retrospective Analysis

**DOI:** 10.3390/ijerph18073583

**Published:** 2021-03-30

**Authors:** Min Xu, Chunxiang Cao, Xin Zhang, Hui Lin, Zhong Yao, Shaobo Zhong, Zhibin Huang, Robert Shea Duerler

**Affiliations:** 1State Key Laboratory of Remote Sensing Science, Aerospace Information Research Institute, Chinese Academy of Sciences, Beijing 100101, China; xumin@radi.ac.cn (M.X.); caocx@aircas.ac.cn (C.C.); zhangxin200353@aircas.ac.cn (X.Z.); huangzhibin18@mails.ucas.ac.cn (Z.H.); duerler2@mails.ucas.ac.cn (R.S.D.); 2School of Geography, Geomatics and Planning, Jiangsu Normal University, Xuzhou 221116, China; 3China Electronic Technology Group Corporation, Institute of Electronic Science, Beijing 100041, China; linhui@cie-info.org.cn; 4Jiangxi Academy of Sciences, Nanchang 330098, China; 5Beijing Research Center of Urban Systems Engineering, Xizhimen Nan Da Jie 16, Xicheng District, Beijing 100035, China; zhongshaobo@gmail.com

**Keywords:** COVID-19, GIS, space-time cluster, retrospective analysis, fine-scale

## Abstract

Exploring spatio-temporal patterns of disease incidence can help to identify areas of significantly elevated or decreased risk, providing potential etiologic clues. The study uses the retrospective analysis of space-time scan statistic to detect the clusters of COVID-19 in mainland China with a different maximum clustering radius at the family-level based on case dates of onset. The results show that the detected clusters vary with the clustering radius. Forty-three space-time clusters were detected with a maximum clustering radius of 100 km and 88 clusters with a maximum clustering radius of 10 km from 2 December 2019 to 20 June 2020. Using a smaller clustering radius may identify finer clusters. Hubei has the most clusters regardless of scale. In addition, most of the clusters were generated in February. That indicates China’s COVID-19 epidemic prevention and control strategy is effective, and they have successfully prevented the virus from spreading from Hubei to other provinces over time. Well-developed provinces or cities, which have larger populations and developed transportation networks, are more likely to generate space-time clusters. The analysis based on the data of cases from onset may detect the start times of clusters seven days earlier than similar research based on diagnosis dates. Our analysis of space-time clustering based on the data of cases on the family-level can be reproduced in other countries that are still seriously affected by the epidemic such as the USA, India, and Brazil, thus providing them with more precise signals of clustering.

## 1. Introduction

A novel coronavirus was first reported in China in December 2019 and was named “COVID-19” by the World Health Organization (WHO) [1,2].The COVID-19 was discovered in the capital of Hubei province, a Central China city named Wuhan, where the traffic system is well developed, and spread rapidly to nearly every part of the world causing global pandemic [3]. The coronavirus is a large virus family, which is known to cause Middle East respiratory syndrome (MERS), severe acute respiratory syndrome (SARS), and other serious diseases [4]. The COVID-19 virus is a new strain of coronavirus that has never been found in a human body before. The virus is mainly transmitted by droplets of coughing or sneezing. In view of this mode of transmission, most cases are related to direct contact, mainly during close contact [5,6,7].

Geography Information System (GIS) is an effective means to visualize and analyze spatial characteristics based on epidemic data [8,9,10]. Combined with spatial statistics, GIS can be used to help mitigate the epidemic through scientific information, find spatial correlations with other variables, and identify transmission dynamics and clustering [11,12,13,14].

Chinese regions differ from one another quite to a great degree, especially in terms of population density, meteorology, transmission net, industry, and economy [15,16,17,18,19,20]. Exploring spatio-temporal patterns of disease incidence can help to identify areas of significantly elevated or decreased risk, providing potential etiologic clues [21,22]. Only through mastering the temporal and spatial distribution of the epidemic can we achieve the target of accurate prevention and control. Quantitative research on the spatial and temporal characteristics of the clusters and the internal diffusion of the virus can not only comprehensively and profoundly help to understand the spatio-temporal law and internal mechanism of epidemic transmission, but also reflects the impact of emergency prevention and control measures for the spread of epidemic and serve as scientific evidence for policy adjustment.

However, most of the studies focus on the single analysis of spatial or temporal clustering. Single analysis of spatial clustering cannot reflect the dynamic change of epidemic situation, and single analysis of temporal aggregation can provide only some fuzzy information, namely whether there exist clusters in a certain time. However, the analysis of space-time clustering can not only indicate whether there exist clusters, but also can detect the spatial location and time of duration. Therefore, the detection results of space-time clustering are more conducive to the disease control departments taking timely response measures. If an outbreak causes the incidence of the whole area to increase at the same time, the temporal clustering method can easily detect the existence of clustering. However, for an outbreak that started in a local area and then gradually spread to the whole area, the incidence curve of the whole area will show an upward trend due to the increase in number of cases in the local area, so the clusters will be detected by single temporal clustering with a lag. The warning results of space-time clustering are more accurate and timelier due to the full use of the temporal and spatial information in the data. It is of great importance for early warning and prevention of future outbreaks [23,24,25].

Currently, epidemiologists pay more attention to the identifying of space-time clusters on a small scale, such as communities, counties, or provinces, and there are very few research works on large-scale analysis of space-time clustering, especially at the national scale [26] Even where there is, due to the availability of data, many studies are conducted with counties or provinces as the smallest unit [27,28]. In fact, it is very important to analyze the clustering of an epidemic on a national scale at the unit of family-level because it is of great significance for the development of accurate prevention, control, and work resumption policy at the national level.

Although we can roughly observe the clustering areas of COVID-19according to the existing epidemic map, the results of clusters number, clustering sizes, and time of duration obtained by different clustering methods at different spatial and temporal scales are very different. Currently, most of the national scale epidemic clustering detection is based on the administrative boundaries of counties, cities, or provinces. However, in the process of infectious disease transmission, the separation of administrative divisions cannot become the barrier of disease transmission. If the detection is carried out in isolation with provinces, cities, and counties as the units, some important clusters may not be detected timely due to the lack of information in the surrounding areas. Therefore, it is necessary to take the family as the smallest unit of clusters detection.

## 2. Materials and Methods

### 2.1. Materials

The COVID-19 cases are collected from the database of diagnosis and suspected cases of COVID-19 in mainland China established by a special group of big data analysis, which is subordinate to the Joint Prevention and Control Mechanism of the State Council. The database is generated based on China’s National Infectious Disease Information System (IDIS), which requires each COVID-19 case to be reported electronically by the responsible doctor as soon as a case has been diagnosed. It includes cases that are reported as asymptomatic, and data are updated in real time. The dataset includes the records of all confirmed COVID-19 cases from the onset of the outbreak on 2 December 2019 to 20 June 2020 in Mainland China (excluding Hong Kong, Macao, and Taiwan regions). Each record contains the information of the patient’s name, gender, ID, date of onset, date of diagnosis, administrative code, home address, and so on. The administrative boundary map of China was acquired from National Geomatics Center of China (http://ngcc.sbsm.gov.cn/, accessed date: 3 March 2018), which is in the format of shapefile.

The daily number of new cases of COVID-19 in mainland China between 2 December 2019 and 20 June 2020 was the data used for Figure 1. The total number of COVID-19 cases is 83,377 as of 20 June 2020. There are two peaks in the variation curve. One is on 24 January 2020 with a count of 3756 confirmed cases. The other one is on 1 February 2020 with a count of 5089 cases. That is because the prevention and control plan of COVID-19 (5th edition) issued by General Office for National Health Commission of China has added “clinical diagnosis” in the case diagnosis classification of Hubei province so that the patients can be diagnosed as early as possible according to the epidemic characteristics of Hubei on 13 February 2020, which led to a surge in new cases near 1 February 2020 as the date of onset.

### 2.2. Methodology

#### 2.2.1. Spatialization of the Case Date

The traditional method of spatialization for epidemic data is based on the administrative code of the patient’s home on county, city, or province scale. However, this method cannot catch the exact location of the case. In this paper, we employ the software of XGeocoding v2 to translate the patient’s home address into coordinate information. In this way, all the recorded confirmed cases were inputted into Microsoft Excel 2010 (Microsoft, Redwoods, WA, USA), geo-coded according to their residential addresses and determined its longitude and latitude coordinates by using ArcGIS10.2 (ESRI Inc., Redlands, CA, USA), which was assumed to represent the location of case outbreak.

#### 2.2.2. Space-Time Cluster Detection

The method employed for cluster detection is space-time scan statistic. Scan statistic is a method widely used in epidemic clustering analysis. It can effectively detect the increase of local time and/or spatial incidence of cases, and test whether the increase is caused by random variation. It can not only detect whether there is clustering in a certain area, but also accurately locate the clustering.

Space-time scan statistics is an extension of spatial scan statistics put forward by Kulldorf, professor of Harvard Medical School in 1997 [29,30,31]. It adds the time dimension to the original spatial scanning statistics so that the scanning statistics can detect the clustering in time and space at the same time. Therefore, compared with the circular window of the spatial scanning statistics, its scanning window is also correspondingly changed into a cylinder, where the bottom of the cylinder corresponds to the spatial range, while the height corresponds to a certain length of time segment. Because the size and position of the cylinder scanning window are constantly changing, the space-time scan statistics can be used for the time and place of epidemic onset. The size of the point and its scale are analyzed in depth, so as to realize the early identification of epidemic outbreak.

The specific detection process of Space-time Scan Statistic can be divided into four steps. Firstly, select a random spatial point in the study area as the center of the bottom surface of the cylinder scanning window. Then, they gradually increase the radius and height of the bottom surface of the cylinder scanning window. The continuous change of the bottom area of the cylinder corresponds to the change of the geographical area covered by the scanning window, and the continuous change of the height of the cylinder corresponds to the change of the bottom area until reaching the maximum space and time limit of the scan window. All positions of the cylinder scan window in the study area repeat the same scanning process. For each scan window, the expected incidence can be calculated according to the actual number of cases and population. Thirdly, the expected incidence can be calculated according to the number of cases in the scan window and outside the scan window. The log likelihood of the test statistics can be constructed from the actual and expected incidence Ratio; LLR (Log Likelihood Ratio) is used to evaluate the abnormal degree of the number of cases in the log likelihood ratio scanning window. It is necessary to select the window with the largest log likelihood ratio as the window with the highest abnormal degree of the number of cases will produce a large number of scanning windows. Finally, use the method of Monte Carlo simulation to evaluate the statistical significance of the window.

The retrospective analyses of space-time permutation is employed as the probability model of clustering detection. Its principle is as follows.

First of all, assume that the number of infections in an area z during d days is *C_zd_*, which corresponds to a scanning cylinder. Then, the total number of infections *C* in the whole study area during all time segments can be expressed as the following function.
(1)$$C=\underset{z}{\sum }\underset{d}{\sum }C_{zd}$$

Thus, the number of infections per day in each region $\mu_{zd}$ can be described as:(2)$$\mu_{zd}=\frac{(\sum_{z}C_{zd})(\sum_{d}C_{zd})}{C}$$

Therefore, we can calculate the expected number of infections $\mu_{A}$ per scanning window *A* according to the number of infections of each unit $\mu_{zd}$.
(3)$$\mu_{A}=\sum_{(z,d)\in A}\mu_{zd}$$

If the observed number of infections in cylinder *A* is *C_A_*, then *C_A_* obeys the hypergeometric distribution of mean $\mu_{A}$. The probability function of *C_A_* can be calculated as:(4)$$P(C_{A})=\frac{\left(\begin{array}{c} \sum_{z\in A}C_{zd}\\ C_{A} \end{array}\right)\left(\begin{array}{c} C-\sum_{z\in A}C_{zd}\\ \sum_{d\in A}C_{zd}-C_{A} \end{array}\right)}{\left(\begin{array}{c} C\\ \sum_{d\in A}C_{zd} \end{array}\right)}$$

When $\sum_{z\in A}C_{zd}$ and $\sum_{d\in A}C_{zd}$ are very small relative to the total number of infections *C*, *C_A_* approximately obeys the Poisson distribution of mean $\mu_{A}$. Based on this approximation, the generalized likelihood ratio (GLR) is used to measure whether the number of infections in cylinder *A* is abnormal.

In this paper, we show how information about the location and spatial extent of such events can be estimated from the spatial and temporal array of all calls by using the space-time permutation scan statistic.
(5)$$GLR={\left\{\frac{C_{A}}{\mu_{A}}\right\}}^{C_{A}}{\left\{\frac{C-C_{A}}{C-\mu_{A}}\right\}}^{C-C_{A}}$$

The probability that a cluster will form by chance is assigned using Monte Carlo hypothesis testing by employing the likelihood of the statistic in question. For this, the time stamps of data points are shuffled and the statistic is calculated again. The process is then repeated 999 times.

The conventional techniques of space-time scan statistic commonly used the administrative boundary such as province, city, and county to be the minimum spatial unit of detection and use the regional center coordinate as all the cases location, which outbreak in this region [27,28].However, the separation of administrative divisions may become the barrier of disease transmission, because some important clusters may not be detected timely due to the lack of information in the surrounding areas if use the provinces, cities, and counties as the minimum unit of the detection. Our method employed the locations of the patients’ community or family to be the basic statistic unit by translating the patients’ home addresses to coordinate in order to detect the space-time clusters in finer scale.

## 3. Results

All 83,377 of the COVID-19 new cases in mainland China between 2 December 2019 and 20 June 2020 are geocoded and spatialized. The spatial distribution of COVID-19 cases are shown in Figure 2. The COVID-19 cases are distributed throughout all the provinces of China. No province is immune. Most of the cases are located in Hubei province, where Wuhan is its provincial capital. Tibet has only one case, which is the least of any province in China. 

All the geo-coded cases were inputted into the model retrospective analyses of space-time permutation using the software of SatScan V9.6 (Department of Population Medicine, Harvard Medical School and Harvard Pilgrim Health Care Institute, Boston, MA, USA). The space-time permutation scan statistic employed in the study utilizes millions of overlapping cylinders to define the scanning window, each being a possible candidate for an outbreak. It can use different parameter settings of the maximum scan cylinders radius and height to consider the spatio-temporal independence of the clusters. When the outbreak location distance of two cases exceeds the maximum scanning radius or the interval time of them exceeds the scanning height, they are considered to be independent, and will not be detected in the same cluster. In this study, we set the maximum cluster size of the spatial window to a circle with a 100 km radius. The maximum temporal cluster size is seven days. The time precision is one day. Table 1 summarizes the characteristics of the statistically significant space-time clusters of COVID-19 at the family-level with a maximum spatial scanning window size of 100km in mainland China. Figure 3 illustrates the distribution of all the corresponding space-time clusters. According to the statistics, 43 clusters were detected with a *p*-value less than 0.05, in which 10 clusters were identified in Hubei province and 4 clusters were in Hebei province. Inner Mongolia and Shandong province had three clusters in each. Heilongjiang, Gansu, Sichuan, Liaoning, Guangdong, Fujian, and Zhejiang province had two clusters in each. Tianjin, Jilin, Jiangxi, Shaanxi, Guizhou, Anhui, Chungking, Shanghai, and Shanxi province had one cluster in each. All the clusters were detected in the central or eastern provinces. No cluster was detected in the western provinces.

The earliest space-time cluster was identified in Zhejiang province from 15–21 January 2020 with a radius of 88.1 km. The earliest cluster in Hubei province was detected from 19–25 January with a radius of 97.1 km. The third province, autonomous region, or municipality directly under the central government that shows a cluster appear is in Inner Mongolia from 2–4 February 2020 with a radius of 7.9 km. Jiangxi and Anhui are the adjacent provinces of Hubei and detected clusters separately from 2–3 February 2020. There were nine clusters detected in Zhejiang, Chongking, Hebei, Hubei, Shandong with a radius of 0 km. This means the cases of the cluster are located in the same communities, and it is likely to be a cluster of the same family. The last cluster was identified in Hebei province from 10–16 June 2020 with a radius of 68.2 km. The observed number of cases is 171, while the expected one is 1.1. In fact, the last cluster is caused by Beijing’s new confirmed COVID-19 cases related to Xinfadi market. Cases in Hebei province account for only a small proportion. With the maximum spatial scanning window size set as 100 km, this cluster contains both areas in Beijing and Hebei, and the cluster center is located in Hebei.

In order to discover finer-scale clusters, we set the maximum cluster size of the spatial window to a circle with a 10 km radius. The maximum temporal cluster size and time precision are still set as seven days and one day. Table 2 summarizes the characteristics of the statistically significant space-time clusters of COVID-19 at the family-level with a maximum spatial scanning window size of 10 km in mainland China. Figure 4 illustrates the distribution of all the corresponding space-time clusters. As statistics, 88 clusters were detected with a *p*-value less than 0.05. The clusters are distributed in 19 provinces, autonomous regions, or municipalities directly under the central government, as shown in Table 3. The earliest clusters are identified in Hubei from 20–25 January, and the last one is in Beijing from 14–20 June. There are 25 clusters located in Hubei province and 8 clusters were in Shanghai city. Each of Beijing, Guangdong, and Inner Mongolia has seven clusters. These areas including Shanghai, Beijing, and Guangdong are China’s top-three well-developed provinces or cities. They have developed transportation network with Hubei province especially Wuhan city. The large population of trade and migrant flow lead to the virus transmitting from Hubei to these areas more quickly than other provinces.

In order to discover more detailed characteristics of the clusters, we enlarged the clustering maps of both Wuhan and Beijing city, which are reported as the original epidemic areas, respectively, in the first and second waves of COVID-19 in China as shown in Figure 5 and Figure 6.There are 16 clusters identified in Wuhan city. All the clusters are detected in February 2020. These clusters are more uniformly distributed in all areas of Wuhan, regardless of whether in urban or suburban areas. In order to help to understand the detail spatio-temporal structure of the clusters, we colored the points of cases by the cluster according to the time from the first case reported in China. The early clusters formed in the east areas of the Yangtze River during 6–10 February 2020, which is about 10 weeks after the first case was reported, although the early cases were reported in Hankou district, which is in the west of the Yangtze River, as shown in Figure 5.

Although there are seven clusters identified in Beijing city, the clustering times range from February 2020 to June 2020. The clusters in Beijing are mainly distributed in the urban area. Similarly, we colored the points of cases by the cluster, and found the clusters can be divided into two stages. The first stage includes four clusters from February to March that are located in the northern urban area of Beijing, which is 12 to 17 weeks from the first reported case. These clusters belong to the first wave of COVID-19 in China, which originated in Wuhan. The second stage includes three clusters of outbreak in June 2020 and is located in the southern urban area of Beijing, which is 27 to 29 weeks from the first reported case. These clusters belong to the second wave of COVID-19 in China, which may have originated in Xinfadi market, which is a large wholesale market that sells fruits, vegetables, and meat located in Beijing’s Fengtai District and has been caught in the spotlight after new COVID-19 clusters were linked to it in June 2020, as shown in Figure 6.

## 4. Discussion

The study employed the retrospective analyses of space-time permutation to detect the space-time clusters of COVID-19 in mainland China on a fine scale. Based on the data obtained from China’s National Infectious Disease Information System (IDIS), we geo-coded each case and translate the patient’s home address into coordinate information in order to catch the exact location of the case. Former studies on space-time clusters detecting of COVID-19 in a county are mainly at the county-level [27,28]. It is the first time for a study to identify the clusters at the family-level in a large country.

Epidemic diagnosis time can scientifically evaluate and comprehensively reflect the emergency level and physical therapy capacity of a national or local health department. Most studies use COVID-19 case data based on diagnosis time from Johns Hopkins University’s Center for Systems Science and Engineering GIS dashboard to do the retrospective analysis. Although these data are updated daily, the statistics of daily cases are based on the diagnosis date. In fact, the average epidemic diagnosis time for COVID-19 outbreak from early onset to diagnosis is 7.35 days in mainland China [32]. It means the detected clusters start and end times will delay 7.35 days using the same space-time scan model based on the diagnosis dates rather than dates of onset. Therefore, it is more scientific and effective to use the case dates of onset when detecting the space-time clusters of an epidemic.

In order to account for the characteristics of the disease in a small region and to improve the probability of detecting smaller clusters, we set the maximum cluster size of the space-time clusters to two fine scales: One is 100 km and another one is 10 km. On the maximum cluster size of 100 km, we have detected 43 clusters during the study period, 10 of which were located in Hubei province. However, to our surprise, the earliest space-time cluster was identified in Zhejiang province from 15–21 January 2020 with a radius of 88.1 km. This may be explained by there being a large flow of population between Hubei and Zhejiang, especially in late January, with many infected students and migrant workers returning to Zhejiang from Wuhan before the closure of the city on 23 January. It also verified one table result in the former study that identifies a significant number of people who entered Wenzhou from Hubei Province, which explains why this city was the first outside the epicenter where confinement was adopted. While the maximum cluster size of 10 km, 88 clusters were detected by space-time scan statistic. The detected clusters are finer than that of 100 km. We have compared the characteristic of the clusters in the city of Wuhan with Beijing. The clusters identified in Wuhan are all detected in February and they are uniformly distributed in all areas of Wuhan, regardless of whether in urban or suburban areas. However, the seven detected clusters in Beijing are mainly distributed in the urban area. Four clusters ranging from February to March are located in the northern urban area. The other three clusters from the outbreak in June are located in southern urban area of Beijing, which may have originated in Xinfadi market and belong to the second wave of COVID-19 in China.

Regardless of whether it isa maximum clustering size of 100 km or 10 km scale, the province where the most clusters are located is Hubei, and the month with the most clusters is February, which indicates that China’s COVID-19 epidemic prevention and control strategy is effective and has successfully prevented the virus from spreading from Hubei to other provinces and lasting too long. When the first wave of the pandemic hit, the virus-testing capabilities were not strong, as it was unbeknownst to us at the time. However, China did a satisfactory job in data tracking, patient tracking, community quarantine, and the early warning from front-line fever clinics, which ensured there were no loopholes left. China was under great pressure when Wuhan city was closed on 23 January 2019. That strategy relieved the situation that the epidemic might transmit into the other areas of China through the Spring Festival holiday and formed more clusters.

There are also some limitations in the study. Firstly, because we use the home coordinate as the unit of space-time scan statistic, no population data at the home address-level can be collected in China. Therefore, we can only select the space-time permutation rather than the Poisson model in the discrete scan statistics. Secondly, the clusters detected are circular. In fact, changes in geography and cultural practices will in many cases invalidate this. Non-circular clusters may help to improve the detection. Thirdly, it is difficult to present all the fine space-time clusters in such a large country as China, especially when there are 88 clusters with a maximum cluster size of 10 km to be exhibited on the map. Finally, because of the change of statistical standards, previously only the number of people who had been diagnosed by accounting instruments were counted as confirmed cases. However, from 12 February 2020, clinical diagnosis cases recognized by doctors were also included in the statistics of confirmed cases, thus causing a sharp increase of cases and clusters in early February for a week from early onset to diagnosis. Lastly, using the conventional space-time scan statistic to detect the fine-scale space-time clusters at the family-level in whole China takes a lot of time. Combined with some of the optimization algorithms such as the particle swarm optimization [33], the probabilistic cellular automata model [34] and the coupled spring forced bat algorithm [35] may help improve the efficiency of the method.

## 5. Conclusions

In this study, we use the retrospective analysis of space-time scan statistic to detect clusters of COVID-19 in mainland China on two fine scales: With maximum clustering radii of 100 km and 10 km. Different from the other study, our analysis is based on case dates of onset, which are collected from the database of diagnosis and suspected cases of COVID-19 in mainland China established by the special group of big data analysis, which is subordinate to the Joint Prevention and Control Mechanism of the State Council. In addition, it is the first time to identify the space-time clusters of COVID-19 in a large country at the family-level. The results show that the detected clusters vary with the maximum clustering radius. Forty-three space-time clusters were detected with a maximum clustering radius of 100 km and 88 clusters with a maximum clustering radius of 10 km from 2 December 2019 to 20 June 2020. Using a small clustering radius may identify finer clusters. Hubei has the most clusters regardless of scale. Most of the clusters were generated in February. That indicates China’s COVID-19 epidemic prevention and control strategy is effective and has successfully prevented the virus from spreading from Hubei to other provinces and lasting too long. Well-developed provinces or cities that have large populations and developed transportation network are more likely to generate space-time clusters. The analysis based on the data of cases of onset may detect the start times of clusters seven days earlier than the same research that is based on the diagnosis dates. Our analysis of space-time clustering based on the data of cases of onset on the family-level can be reproduced in other countries that are still seriously affected by the epidemic, such as the USA, India, and Brazil, providing them with more precise signals of clustering.

## Figures and Tables

**Figure 1 ijerph-18-03583-f001:**
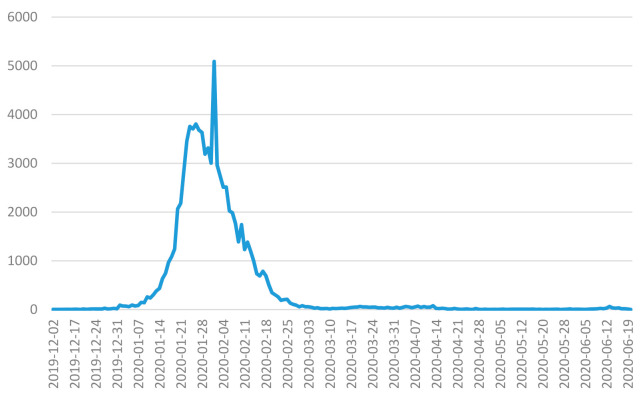
Daily number of COVID-19 cases in mainland China between 2 December 2019 and 20 June 2020 (dates of onset are used for the statistics).

**Figure 2 ijerph-18-03583-f002:**
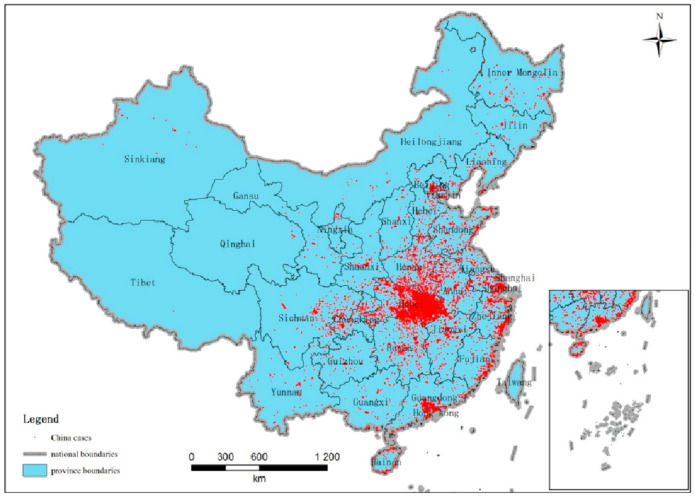
The spatial distribution of COVID-19 in mainland China between 2 December 2019 and 20 June 2020 (Taiwan, Hang Kong and Macau not included).

**Figure 3 ijerph-18-03583-f003:**
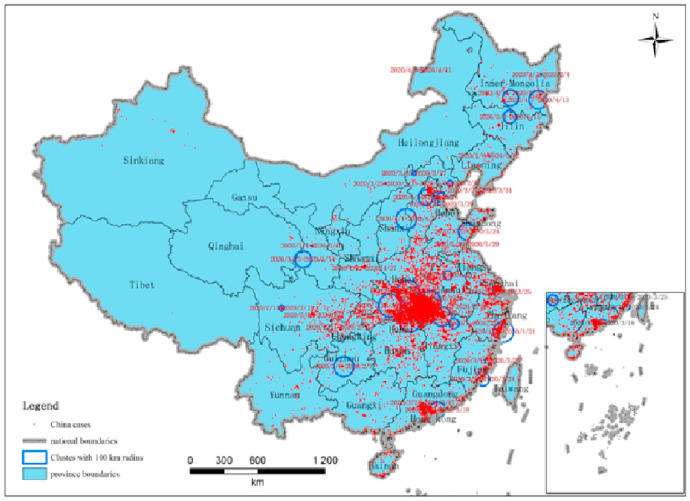
Space-time clusters of COVID-19 at the family-level with a maximum spatial scanning window size of 100 km in mainland China from 2 December 2019 to 20 June 2020.

**Figure 4 ijerph-18-03583-f004:**
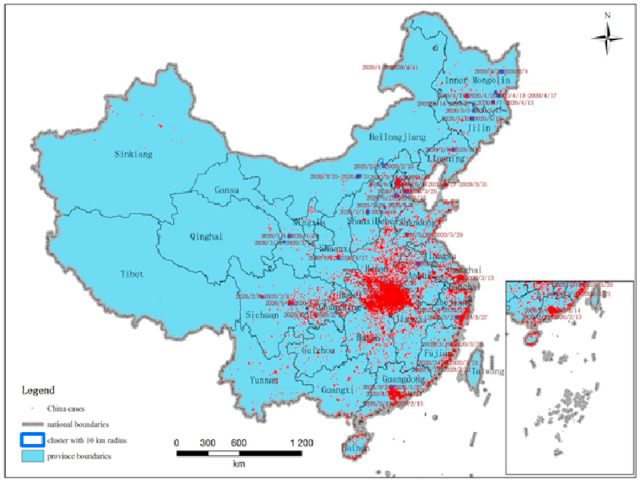
Space-time clusters of COVID-19 at the family-level with a maximum spatial scanning window size of 10 km in mainland China from 2 December 2019 to 20 June 2020.

**Figure 5 ijerph-18-03583-f005:**
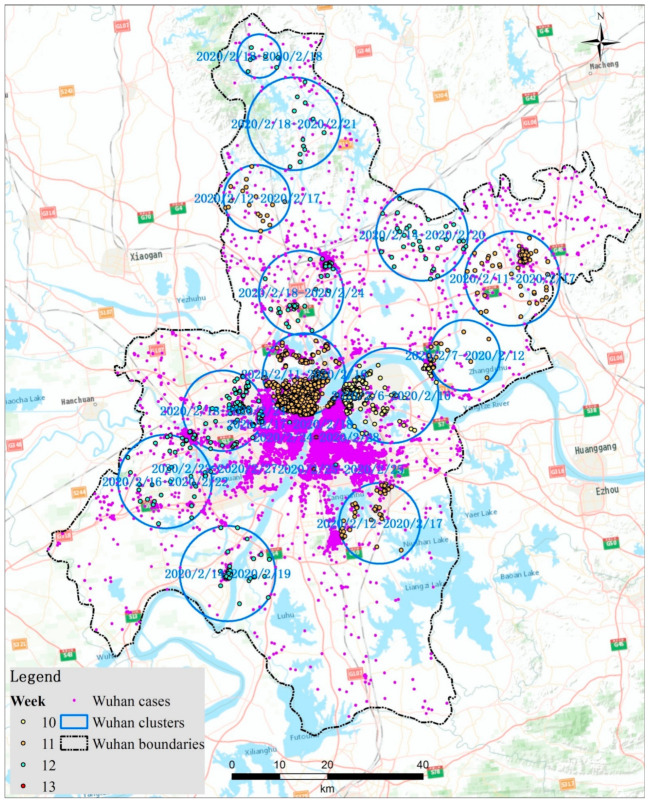
Zoomed space-time clusters of COVID-19 at the family-level with a maximum spatial scanning window size of 10 km in Wuhancity from 2 December 2019 to 20 June 2020.

**Figure 6 ijerph-18-03583-f006:**
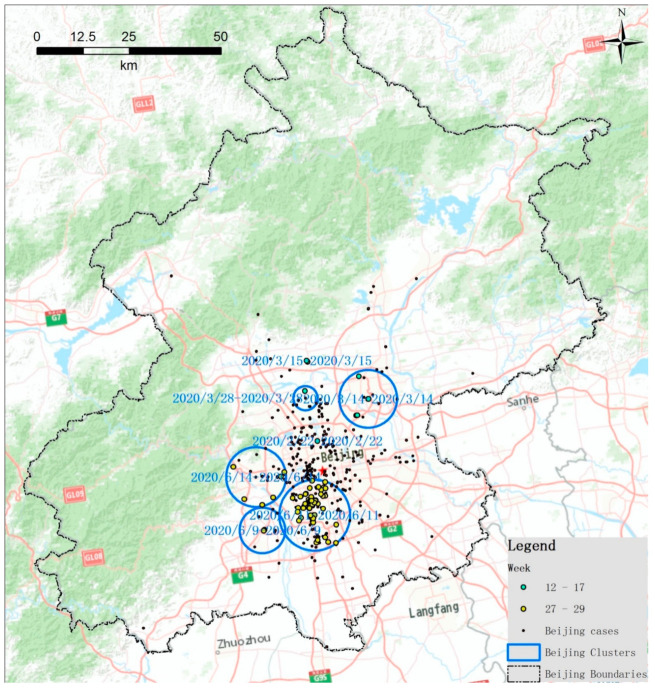
Zoomed space-time clusters of COVID-19 at the family-level with a maximum spatial scanning window size of 10 km in Beijing city from 2 December 2019 to 20 June 2020.

**Table 1 ijerph-18-03583-t001:** Characteristics of the statistically significant space-time clusters of COVID-19 at the family-level with a maximum spatial scanning window size of 100 km in mainland China from 2 December 2019 to 20 June 2020 (clusters where all cases happen in the same geolocation are reported as having 0 km radii).

SN	Cluster Center	Radius (km)	Start Date	End Date	Temporal Size (d)	*p*-Value	Observed	Expected
1	Zhejiang	88.1	15 January 2020	21 January 2020	7	<0.001	130	60.35
2	Hubei	97.1	19 January 2020	25 January 2020	7	<0.001	933	674.71
3	Hubei	100.0	22 January 2020	28 January 2020	7	<0.001	487	341.09
4	Hubei	83.2	23 January 2020	27 January 2020	5	<0.001	353	237.13
5	Hubei	99.7	28 January 2020	31 January 2020	4	<0.001	670	378.61
6	Inner Mongolia	7.9	2 February 2020	4 February 2020	3	<0.001	30	4.33
7	Jiangxi	43.7	2 February 2020	5 February 2020	4	0.005	33	8.62
8	Anhui	39.4	3 February 2020	7 February 2020	5	<0.001	64	22.88
9	Hubei	26.8	3 February 2020	8 February 2020	6	0.002	64	25.17
10	Guizhou	85.2	6 February 2020	7 February 2020	2	0.001	20	2.94
11	Hebei	24.8	8 February 2020	14 February 2020	7	0.001	24	4.19
12	Zhejiang	0.0	9 February 2020	11 February 2020	3	<0.001	20	1.94
13	Chungking	0.0	11 February 2020	11 February 2020	1	<0.001	7	0.10
14	Hebei	0.0	11 February 2020	11 February 2020	1	0.040	7	0.21
15	Hubei	86.5	14 February 2020	20 February 2020	7	<0.001	712	369.38
16	Sichuan	22.5	15 February 2020	19 February 2020	5	<0.001	25	2.94
17	Hubei	0.0	17 February 2020	18 February 2020	2	<0.001	311	11.86
18	Hubei	0.0	19 February 2020	21 February 2020	3	<0.001	85	1.18
19	Shandong	0.0	19 February 2020	19 February 2020	1	<0.001	122	1.25
20	Hubei	0.0	24 February 2020	28 February 2020	5	<0.001	25	0.29
21	Hubei	0.0	25 February 2020	25 February 2020	1	<0.001	23	0.13
22	Hebei	17.6	25 February 2020	27 February 2020	3	0.009	5	0.04
23	Gansu	6.3	4 March 2020	10 March 2020	7	<0.001	35	0.14
24	Guangdong	65.5	12 March 2020	18 March 2020	7	<0.001	14	1.04
25	Gansu	73.7	13 March 2020	14 March 2020	2	<0.001	5	0.01
26	Fujian	57.6	18 March 2020	24 March 2020	7	<0.001	24	0.46
27	Shanghai	53.6	19 March 2020	25 March 2020	7	<0.001	103	2.64
28	Fujian	30.9	19 March 2020	25 March 2020	7	<0.001	12	0.28
29	Guangdong	29.2	21 March 2020	27 March 2020	7	<0.001	51	2.05
30	Shandong	76.4	22 March 2020	24 March 2020	3	0.007	7	0.16
31	Tianjin	41.5	23 March 2020	29 March 2020	7	<0.001	27	0.43
32	Heilongjiang	7.3	25 March 2020	31 March 2020	7	<0.001	33	0.24
33	Liaoning	8.8	27 March 2020	31 March 2020	5	<0.001	7	0.03
34	Shanxi	92.2	1 April 2020	7 April 2020	7	<0.001	44	0.43
35	Heilongjiang	20.0	5 April 2020	11 April 2020	7	<0.001	59	0.35
36	Inner Mongolia	82.8	7 April 2020	13 April 2020	7	<0.001	238	2.11
37	Inner Mongolia	72.2	14 April 2020	20 April 2020	7	<0.001	32	0.32
38	Shaanxi	3.6	21 April 2020	27 April 2020	7	<0.001	33	0.05
39	Liaoning	5.4	8 May 2020	12 May 2020	5	0.003	3	<0.01
40	Jilin	60.0	9 May 2020	15 May 2020	7	<0.001	23	0.03
41	Sichuan	0.8	28 May 2020	1 June 2020	5	<0.001	11	0.01
42	Shandong	0.0	29 May 2020	29 May 2020	1	<0.001	3	<0.01
43	Hebei	68.2	10 June 2020	16 June 2020	7	<0.001	171	1.10

**Table 2 ijerph-18-03583-t002:** Characteristics of the statistically significant space-time clusters of COVID-19 at the family-level with a maximum spatial scanning window size of 10 km in mainland China from 2 December 2019 to 20 June 2020 (clusters where all cases happen in the same geo-location are reported as having 0 km radii).

SN	Cluster Center	Latitude	Longitude	Radius	Start Time	End Time	Temporal Size (d)	*p*-Value	Observed	Expected
1	Hubei	30.9662	113.9902	8.3	20 January 2020	25 January 2020	6	0.005	163	95.3
2	Hubei	31.6174	113.8294	9.4	25 January 2020	31 January 2020	7	0	132	64.8
3	Hubei	32.0496	112.1332	5.2	26 January 2020	31 January 2020	6	0	188	101.4
4	Hubei	31.6887	113.4446	8.9	29 January 2020	31 January 2020	3	0	153	54.1
5	Hubei	30.7732	112.6343	7.7	3 February 2020	7 February 2020	5	0	36	8.3
6	Hubei	30.4947	113.5432	2.8	3 February 2020	4 February 2020	2	0	23	3.4
7	Hubei	30.6153	114.4855	10	6 February 2020	10 February 2020	5	0	564	378.2
8	Hubei	30.674	114.6567	7.4	7 February 2020	12 February 2020	6	0.005	45	14.8
9	Sichuan	31.0197	101.1555	1.2	8 February 2020	8 February 2020	1	0	12	0.5
10	Zhejiang	29.0978	119.0652	0	9 February 2020	11 February 2020	3	0	20	1.9
11	Hubei	30.6753	114.3019	8.6	11 February 2020	16 February 2020	6	0	1255	806.7
12	Hubei	30.807	114.7813	9.9	11 February 2020	17 February 2020	7	0	78	30.1
13	Chungking	29.6338	106.3409	0	11 February 2020	11 February 2020	1	0	7	0.1
14	Hebei	41.7145	114.7829	0	11 February 2020	11 February 2020	1	0.027	7	0.2
15	Hubei	30.3781	114.4275	8.5	12 February 2020	17 February 2020	6	0	69	19.9
16	Hubei	31.0169	114.2449	7	12 February 2020	17 February 2020	6	0	31	6.5
17	Guangdong	22.1659	113.2833	8.1	13 February 2020	13 February 2020	1	0.002	8	0.2
18	Inner Mongolia	44.9239	127.177	0	14 February 2020	15 February 2020	2	0.011	7	0.2
19	Hubei	30.3177	114.0855	10	14 February 2020	19 February 2020	6	0	98	33.4
20	Hubei	30.9102	114.5958	9.6	14 February 2020	20 February 2020	7	0	50	12.3
21	Hubei	30.4973	114.89	9.2	14 February 2020	14 February 2020	1	0	45	10.7
22	Hubei	30.504	113.9726	9.8	16 February 2020	21 February 2020	7	0	52	15.8
23	Hubei	30.5878	114.2588	0	17 February 2020	18 February 2020	2	0	311	11.9
24	Hubei	30.6244	114.1162	8.5	18 February 2020	24 February 2020	7	0	172	64.4
25	Hubei	31.2838	114.2869	4.6	18 February 2020	18 February 2020	1	0	19	0.3
26	Hubei	30.8288	114.317	8.9	18 February 2020	24 February 2020	7	0	66	15.4
27	Hubei	31.1484	114.3455	9.8	18 February 2020	21 February 2020	4	0	21	2
28	Hubei	30.4977	114.3479	0	19 February 2020	21 February 2020	3	0	85	1.2
29	Shandong	35.5204	116.5637	0	19 February 2020	19 February 2020	1	0	122	1.2
30	Inner Mongolia	46.7797	131.8194	7.9	2 February 2020	4 February 2020	3	0	30	4.3
31	Anhui	32.9803	117.3249	7.9	2 February 2020	7 February 2020	6	0.022	56	22.1
32	Beijing	39.9807	116.3938	0	22 February 2020	26 February 2020	5	0	10	0.1
33	Hubei	30.515	114.0829	0.8	22 February 2020	27 February 2020	6	0	24	0.4
34	Hubei	30.554	114.3122	0	24 February 2020	28 February 2020	5	0	25	0.3
35	Hubei	30.4946	114.2993	0	25 February 2020	25 February 2020	1	0	23	0.1
36	Hebei	41.4276	114.9566	2.2	25 February 2020	25 February 2020	1	0.018	4	0
37	Gansu	36.0742	103.7648	6.3	4 March 2020	10 March 2020	7	0	35	0.1
38	Guangdong	22.5322	113.9951	6.9	8 March 2020	12 March 2020	5	0	8	0.1
39	Guangdong	22.9355	113.4241	9.5	8 March 2020	14 March 2020	7	0	7	0.1
40	Shanghai	31.1188	121.6558	9.9	10 March 2020	13 March 2020	4	0	10	0.1
41	Gansu	35.6057	103.2067	0	13 March 2020	14 March 2020	2	0	5	0
42	Beijing	40.0674	116.5792	8.1	14 March 2020	20 March 2020	7	0	61	0.4
43	Beijing	40.1854	116.3973	0.4	15 March 2020	21 March 2020	7	0	17	0.1
44	Fujian	24.5038	118.0441	7.1	15 March 2020	21 March 2020	7	0	9	0.1
45	Fujian	24.724	118.7083	7.8	16 March 2020	21 March 2020	6	0	9	0
46	Shanghai	31.3344	121.6008	9.9	18 March 2020	24 March 2020	7	0.011	7	0.2
47	Tianjin	39.0995	117.2372	9.7	19 March 2020	25 March 2020	7	0	16	0.3
48	Shanghai	31.2015	121.4778	9.4	19 March 2020	25 March 2020	7	0	50	0.9
49	Fujian	26.0801	119.3766	8.4	19 March 2020	25 March 2020	7	0.022	6	0.1
50	Shanghai	31.0122	121.4134	0.1	20 March 2020	25 March 2020	6	0.001	5	0
51	Sichuan	30.5619	103.9225	3.1	21 March 2020	26 March 2020	6	0	5	0
52	Guangdong	23.1856	113.3322	9.3	21 March 2020	27 March 2020	7	0	32	0.8
53	Heilongjiang	40.9065	111.9804	0	22 March 2020	22 March 2020	1	0	6	0
54	Zhejiang	28.0965	120.3317	8.7	22 March 2020	27 March 2020	6	0.013	5	0
55	Tianjin	39.0041	117.7655	0	23 March 2020	28 March 2020	6	0	8	0
56	Heilongjiang	40.8446	111.7303	7.3	25 March 2020	31 March 2020	7	0	33	0.2
57	Hebei	39.3359	117.8391	2.8	27 March 2020	29 March 2020	3	0	4	0
58	Liaoning	38.9591	121.6183	8.8	27 March 2020	31 March 2020	5	0	7	0
59	Beijing	40.092	116.3751	3.4	28 March 2020	29 March 2020	2	0	5	0
60	Hebei	38.0112	114.5161	0	28 March 2020	3 April 2020	7	0	6	0
61	Guangdong	23.4414	113.3113	8.4	29 March 2020	4 April 2020	7	0.022	5	0
62	Shanxi	37.7926	112.5359	8.1	31 March 2020	6 April 2020	7	0	29	0.2
63	Shanghai	30.8993	121.165	0	1 April 2020	4 April 2020	4	0.003	4	0
64	Shanghai	31.0648	121.7437	0	3 April 2020	9 April 2020	7	0	31	0.3
65	Shanghai	31.2074	121.7187	0	3 April 2020	9 April 2020	7	0	16	0.1
66	Heilongjiang	49.48	117.6884	0	4 April 2020	10 April 2020	7	0	30	0.1
67	Heilongjiang	49.6056	117.4898	2.9	5 April 2020	11 April 2020	7	0	31	0.2
68	Inner Mongolia	44.6009	129.6144	2.2	6 April 2020	12 April 2020	7	0	25	0.2
69	Shanxi	37.9272	112.5666	3	7 April 2020	7 April 2020	1	0	12	0
70	Inner Mongolia	44.4032	131.1704	5.1	7 April 2020	13 April 2020	7	0	208	1.6
71	Inner Mongolia	44.9222	130.5428	4.9	13 April 2020	17 April 2020	5	0	9	0
72	Inner Mongolia	45.7625	126.6578	6.2	14 April 2020	20 April 2020	7	0	28	0.2
73	Tianjin	39.1706	117.3875	1.3	16 April 2020	16 April 2020	1	0.003	3	0
74	Shaanxi	34.2054	108.9867	3.6	21 April 2020	27 April 2020	7	0	33	0.1
75	Inner Mongolia	45.2905	130.2865	3.3	25 April 2020	30 April 2020	6	0	4	0
76	Jilin	44.4123	126.9714	1.6	5 May 2020	11 May 2020	7	0	11	0
77	Liaoning	41.7046	123.4162	5.4	8 May 2020	12 May 2020	5	0.002	3	0
78	Jilin	43.8046	126.532	8	10 May 2020	15 May 2020	6	0	17	0
79	Shanghai	31.339	121.4488	0	13 May 2020	13 May 2020	1	0	4	0
80	Sichuan	30.6483	104.0582	0.8	28 May 2020	1 June 2020	5	0	11	0
81	Shandong	35.1304	119.2983	0	29 May 2020	29 May 2020	1	0	3	0
82	Guangdong	23.0409	113.2594	6.5	31 May 2020	6 June 2020	7	0	5	0
83	Sichuan	30.5971	104.1595	5.9	3 June 2020	8 June 2020	6	0	5	0
84	Guangdong	23.1688	113.4528	1.5	8 June 2020	13 June 2020	6	0	10	0
85	Beijing	39.7775	116.1731	6.4	9 June 2020	12 June 2020	4	0.024	4	0
86	Hebei	38.8718	115.9767	9.3	10 June 2020	15 June 2020	6	0	7	0
87	Beijing	39.7966	116.3498	9.9	11 June 2020	17 June 2020	7	0	144	0.7
88	Beijing	39.9131	116.1772	8.4	14 June 2020	20 June 2020	7	0	13	0.1

**Table 3 ijerph-18-03583-t003:** Number of clusters in each province at the family-level with a maximum spatial scanning window size of 10 km.

Province Name	Number of Clusters
Anhui	1
Beijing	7
ChongKing	1
Fujian	3
Gansu	1
Guangdong	7
Hebei	5
Heilongjiang	4
Hubei	25
Inner Mongolia	7
Jilin	2
Liaoning	2
Shaanxi	1
Shandong	2
Shanghai	8
Shanxi	2
Sichuan	4
Tianjin	3
Zhejiang	2

## Data Availability

No additional data available.

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
