# Peer review of "Fine-Scale Space-Time Cluster Detection of COVID-19 in Mainland China Using Retrospective Analysis"

_ijerph, 2021, doi:10.3390/ijerph18073583_

Round 1

Reviewer 1 Report

Dear authors 

The article is prepared really well. I just highly recommend you to work more on the discussion part. It can be more interesting for the authors to follow how do you want to use the results. 

Author Response

Point 1: The article is prepared really well. I just highly recommend you to work more on the discussion part. It can be more interesting for the authors to follow how do you want to use the results. 

Response 1: Thanks to the reviewer’s suggestion. We have added some more discussions on the discussion part according to the comment.

Reviewer 2 Report

The paper presents a method of clustering spatio-temporal data of COVID-19 cases with a retrospective approach. The approach is interesting, especially in the public health context owing to the probability model of infections. The results of a case study for China mainland were quite fit to the actual trends of COVID-19 in the past. As the authors suggest, the approach is useful to identify spatial trends of the infections in finer scales so that the authorities can address countermeasures in earlier stages. I would like to recommend this manuscript for publication in the journal because of the practical usefulness in the developing issue of COVID-19.

Before the publication, I would like the authors to address my minor comment below:

1. Figure 5 - Please consider coloring the points by the cluster. It will help to understand in how the method is useful to depict detail spatio-temporal structure more than just circles.

2. Please insert the figures with vector graphics or high-resolution images. The current figures are hard to read.

Author Response

Point 1: Figure 5 - Please consider coloring the points by the cluster. It will help to understand in how the method is useful to depict detail spatio-temporal structure more than just circles. 

Response 1: Thanks to the reviewer’s suggestion. we have colored the points of cases by the cluster according to the time from the first case reported in China in order to help to understand the detail spatio-temporal structure of the clusters, as shown in figure 5 and figure 6.

Point 2: Please insert the figures with vector graphics or high-resolution images. The current figures are hard to read.

Response 2: We have insert higher resolution images in the manuscript according to the reviewer’s comments, as shown in figure 2-6.

Reviewer 3 Report

In this work, the authors propose to detect spatio-temporal clusters of COVID-19 according to a retrospective approach. One of the key points of this work is to detect clusters at a fine spatial scale of the order of the family unit. Indeed, the problem considered here is very important in the context of COVID-19 where the transmission scales are from very local to broad.

However, there are major points:

- the method developed and used is not easily understood. The document would benefit from providing more description and comparison with conventional techniques (which should also be cited)

 - most of these clustering methods suppose a spatio-temporal independence of the clusters; which is clearly not the case with COVID-19 because there are obviously contagion processes. How is this taken into account in this approach?

- It would also be interesting to provide the distributions of the spatial and temporal sizes of the identified clusters.

- How are the clusters organized or structured? Is there a hierarchy of clusters or groupings of clusters into super clusters?

The paper reports some interesting results that can be published. However, I recommend that authors to rework the manuscript by addressing carefully the point mentioned above before consideration of publication.

Author Response

Point 1: The method developed and used is not easily understood. The document would benefit from providing more description and comparison with conventional techniques (which should also be cited). 

Response 1: Thanks to the reviewer’s comment. We have added some description and comparison with conventional techniques in the last paragraph of the Methods section: “The conventional techniques of space-time scan statistic commonly used the administrative boundary such as province, city and county to be the minimum spatial unit of detection and use the regional center coordinate as all the cases location which outbreak in this region[27,28]. But the separation of administrative divisions may become the barrier of disease transmission, because some important clusters may not be detected timely due to the lack of information in the surrounding areas if use the provinces, cities and counties as the minimum unit of the detection. Our method employed the locations of the patients’ community or family to be the basic statistic unit by translating the patients’ home addresses to coordinate in order to detect the space-time clusters in finer scale.”

Point 2: most of these clustering methods suppose a spatio-temporal independence of the clusters; which is clearly not the case with COVID-19 because there are obviously contagion processes. How is this taken into account in this approach?

Response 2: The space–time permutation scan statistic employed in the paper utilizes thousands or millions of overlapping cylinders to define the scanning window, each being a possible candidate for an outbreak. It can use different parameter settings of the maximum scan cylinders radius and height to consider the spatio-temporal independence of the clusters. When the outbreak location distance of two cases exceeds the maximum scanning radius or the interval time of them exceeds the scanning height, they are considered to be independent, and won’t be detected in the same cluster. In this study, we set the maximum cluster size of the cylinders with a 100 km or 10 km radius. The maximum temporal cluster size is set as 7 days. We have added these descriptions about spatio-temporal independence of the clusters in the second paragraph of the results section.

Point 3: It would also be interesting to provide the distributions of the spatial and temporal sizes of the identified clusters.

Response 3: We have added the spatial and temporal sizes of the identified clusters in table 1 and table 2 in the revised manuscript according to reviewer’s comment. The field of Radius(km) is the spatial size of the clusters. The temporal size is calculated by the fields of start time and end time.

Point 4: How are the clusters organized or structured? Is there a hierarchy of clusters or groupings of clusters into super clusters?

Response 2: There is no hierarchy of clusters or groupings of clusters into super clusters. Because the method requires a setting of the maximum scan cylinders radius and height. The circular base represents the geographical area of the potential outbreak. A typical approach is to first iterate over a finite number geographical grid points and then gradually increase the circle radius from zero to some maximum value defined by the user, iterating over the coordinates in the order in which they enter the circle. In this way, both small and large circles are considered, all of which overlap with many other circles. The height of the cylinder represents the number of days, with the requirement that the last day is always included together with a variable number of preceding days, up to some maximum defined by the user. For example, we may consider all cylinders with a height of 7 d. For each center and radius of the circular cylinder base, the method iterates over all possible temporal cylinder lengths, and may formed at most one cluster in each centre.

Reviewer 4 Report

1, the equations in the paper should be rewritten. The size is not very suitable to the whole presentation of the paper.

2, the figures in the paper should be improved. Current version is very difficult to read

3, space-time cluster detection was proposed around 1997. Did the authors check the improvement of this method after that? Why not consider some recent results of this method?

4, as far as I concerned, some of the optimization algorithms may help improve this method, can the authors provide some future discussions?  For example:

[1] He, Shaobo, Yuexi Peng, and Kehui Sun. "SEIR modeling of the COVID-19 and its dynamics." Nonlinear Dynamics 101.3 (2020): 1667-1680.

[2] Ghosh, Sayantari, and Saumik Bhattacharya. "A data-driven understanding of COVID-19 dynamics using sequential genetic algorithm based probabilistic cellular automata." Applied Soft Computing 96 (2020): 106692.

[3] Zhang, Haopeng, and Qing Hui. "A Coupled Spring Forced Bat Searching Algorithm: Design, Analysis and Evaluation." 2020 American Control Conference (ACC). IEEE, 2020.

Author Response

Point 1: the equations in the paper should be rewritten. The size is not very suitable to the whole presentation of the paper. 

Response 1: Thanks to the reviewer’s suggestion. we have rewritten all the equations.

Point 2: the figures in the paper should be improved. Current version is very difficult to read

Response 2: We have redrawn the figure 2-6 and insert higher resolution images in the revised manuscript according to the comments.

Point 3: space-time cluster detection was proposed around 1997. Did the authors check the improvement of this method after that? Why not consider some recent results of this method?

Response3: The original spatial scan statistics was purposed in 1997. But the conventional space–time permutation scan statistic was purposed in 2005. I have added the cited reference of space–time permutation scan statistic in the revised manuscript. Our method is based on the conventional space–time permutation scan statistic.“The conventional techniques of space-time scan statistic commonly used the administrative boundary such as province, city and county to be the minimum spatial unit of detection and use the regional center coordinate as all the cases location which outbreak in this region[27,28]. But the separation of administrative divisions may become the barrier of disease transmission, because some important clusters may not be detected timely due to the lack of information in the surrounding areas if use the provinces, cities and counties as the minimum unit of the detection. Our method employed the locations of the patients’ community or family to be the basic statistic unit by translating the patients’ home addresses to coordinate in order to detect the space-time clusters in finer scale.

Point 4: as far as I concerned, some of the optimization algorithms may help improve this method, can the authors provide some future discussions? 

Response 4: Thanks to the reviewer’s suggestion. The authors have provide some discussions about the optimization algorithms in the revised manuscript.

Round 2

Reviewer 3 Report

The authors have addressed almost appropriately all my queries.

Minor queries:

  1. Table 2 - report the "p" value in full or as "p < 0.00?"
  2. Edit for english

Reviewer 4 Report

the authors addressed my comments